# BCL2 Protein Progressively Declines during Robust CLL Clonal Expansion: Potential Impact on Venetoclax Clinical Efficacy and Insights on Mechanism

Hyunjoo Lee [1], Shabirul Haque [1], Rashmi Gupta [1], Jonathan E. Kolitz [1,2,3], Steven L. Allen [1,2,3], Kanti Rai [1,3,4], Nicholas Chiorazzi [1,2,3,4] and Patricia K. A. Mongini [1,3,*,†]

[1]  The Feinstein Institutes for Medical Research, Northwell Health, Manhasset, NY 11030, USA; shaque@northwell.edu (S.H.); sallen@northwell.edu (S.L.A.); nchizzi@northwell.edu (N.C.)
[2]  Department of Medicine, Zucker School of Medicine at Hofstra/Northwell, Hempstead, NY 11549, USA
[3]  Department of Molecular Medicine, Zucker School of Medicine at Hofstra/Northwell, Hempstead, NY 11549, USA
[4]  Northwell Health Cancer Institute, Lake Success, NY 11042, USA
[*]  Correspondence: patricia.mongini@nau.edu
[†]  Current address: Pathogen and Microbiome Institute, Northern Arizona University, Flagstaff, AZ 86011, USA.

**Abstract:** CLL B cells express elevated pro-survival BCL2, and its selective inhibitor, venetoclax, significantly reduces leukemic cell load, leading to clinical remission. Nonetheless, relapses occur. This study evaluates the hypothesis that progressively diminished BCL2 protein in cycling CLL cells within patient lymph node niches contributes to relapse. Using CFSE-labeled, purified CLL populations known to respond with vigorous cycling in d6 cultures stimulated with TLR9-activating ODN (oligodeoxynucleotide) + IL15, we show that BCL2 protein progressively declines during consecutive cell divisions. In contrast, MCL1 and survivin are maintained/slightly elevated during cycling. Delayed pulsing of quiescent and activated CLL cultures with selective inhibitors of BCL2 or survivin revealed selective targeting of noncycling and cycling populations, respectively, raising implications for therapy. To address the hypothesis that BCL2-repressive miRs (*miR15a/miR16-1*), encoded in Chr13, are mechanistically involved, we compared BCL2 protein levels within ODN + IL15-stimulated CLL cells, with/without del(13q), yielding results suggesting these miRs contribute to BCL2 reduction. In support, within ODN-primed CLL cells, an IL15-driven STAT5/PI-3K pathway (required for vigorous cycling) triggers elevated p53 TF protein known to directly activate the *miR15a/miR16-1* locus. Furthermore, IL15 signaling elicits the repression of *BCL2* mRNA within 24 h. Additional comparisons of del(13q)+ and del(13q)−/− cohorts for elevated p53 TF expression during cycling suggest that a documented *miR15a/miR16-1*-mediated negative feedback loop for p53 synthesis is active during cycling. Findings that robust CLL cycling associates with progressively decreasing BCL2 protein that directly correlates with decreasing venetoclax susceptibility, combined with past findings that these cycling cells have the greatest potential for activation-induced cytosine deaminase (AICDA)-driven mutations, suggest that venetoclax treatment should be accompanied by modalities that selectively target the cycling compartment without eliciting further mutations. The employment of survivin inhibitors might be such an approach.

**Keywords:** CLL; BCL2; p53; survivin; venetoclax; YM155





## 1. Introduction

The survival of both normal and malignant lymphocytes, in large part, reflects an orchestrated balance between proteins promoting viability and those eliciting death. The outcome for transformed cells is typically skewed toward survival due to critical genetic aberrations and/or a supportive milieu.

CLL, the most common adult leukemia in the United States and Europe, typically appears in individuals >60 years of age as a slow accumulation of small CD5+ B lymphocytes in the blood [1,2]. This suggests that leukemia represents the heightened survival of a quiescent clonal population rather than aberrant clonal proliferation. In support of this, the pro-survival BCL2 protein is elevated in CLL [3–5], with the latter often associated with del(13q) [6,7], a chromosomal anomaly that deletes two *BCL2* regulatory micro-RNA (miR), *miR15a* and *miR16-1* [6–9]. Consequently, specific BCL2 inhibitors were developed and tested in clinical trials. Treatment with the selective BCL2 inhibitor, venetoclax [10], is highly effective at achieving remission and improving patient outcomes [11]. Unfortunately, venetoclax treatment alone or in combination with several other agents does not achieve a cure [12].

Explanations for venetoclax's incomplete therapeutic efficacy have been sought. An outgrowth of acquired clonal variants with functional *BCL2* mutations conferring resistance occurs [13–15]. Resistance is also linked to variants with stable epigenetic suppression of pro-apoptotic PUMA [16]; genetic/epigenetic amplification of MCL1, a short-lived pro-survival molecule of the BCL2 family [17,18]; and sometimes, the amplification of pre-existing or therapy-elicited mutations in non-*BCL2* genes affecting BCL2 dependence [19]. Most recently, the hyperphosphorylation of several BCL2 family proteins, including BCL2, MCL1, BAD, and BAX, was linked to resistance in lymphoid malignancies, including CLL [20]. Finally, functional refractoriness to venetoclax has been linked to characteristically elevated expression of alternative pro-survival molecules, MCL1, BCLXL, and survivin, within reservoirs of activated CLL cells (pseudofollicles) in lymphoid tissues [21–24]. Consistent with a known circulation of blood leukemic cells into lymphatic tissues [25], a relatively short-term (24–72 h) culture of circulating CLL cells with stimuli found in LNs, ligands for BCR or CD40, and cytokines elicits significantly elevated MCL1, BCLXL, and survivin (BIRC5) [26–29], with no evidenced change in BCL2 protein during this period [28,29]. Decreased leukemic cell vulnerability to venetoclax accompanies these changes [28,29].

The following study addresses the hypothesis that an additional factor affecting venetoclax resistance is the selective loss of its target, BCL2, within the actively dividing subset of CLL cells in pseudofollicles [30]. There are reasons to suspect this. Division-linked declines in BCL2 are evident in normal human B cells, both within T cell-dependent germinal centers (GC) [31] and within cycling B lymphoblasts elicited in vitro by certain T cell independent stimuli [32]. Furthermore, BCL2 is often reduced in CLL pseudofollicles, as evaluated by immunohistochemistry [21,33].

To test this hypothesis, a well-characterized in vitro surrogate for in vivo CLL growth in pseudofollicles [2,34,35] was employed. CLL clonal expansion was elicited with synergistic stimuli, CpG DNA and IL15, both present within lymphoid tissue pseudofollicles [2,35]. A flow cytometric approach involving CLL cells labeled with a division-tracking fluorescent dye [2] and fluorescent staining of intracellular proteins permitted the quantitative monitoring of BCL2/MCL1/survivin and p53 proteins as activated CLL cells progressed through sequential cell divisions. Additionally, measurements of viability and absolute viable cell yield in cycling CLL cultures pulsed with venetoclax or survivin inhibitor (YM155) [36] permitted insights into the relative susceptibility of division subsets to inhibitor-induced death.

## 2. Results

BCL2 protein levels notably decline within extended cultures of CLL cells receiving proliferation-inducing signals from ODN + IL15. A comparison of BCL2 levels within the CLL clonal population (U-CLL950) before in vitro activation versus after 5d stimulation with synergistic ODN + IL15 revealed a pronounced decline (71%) associated with stimulation (Figure 1A). A similar activation-related decline in BCL2 protein was noted in a separate experiment (Figure 1B) upon comparing BCL2 expression within viable-gated U-CLL1013 cells from unstimulated d6 cultures versus that within parallel ODN + IL15-stimulated

cultures (87% decrease). In contrast, when MCL1 levels were assessed (Figure 1A), no such decrease was noted in ODN + IL15-stimulated U-CLL950 cells. Rather, activated cultures exhibited greater MCL1 than noted prior to culture (36% increase), a finding consistent with past reports of elevated MCL1 protein levels within 24–72 h after CLL activation by diverse stimuli [26–29].

To directly examine whether BCL2 levels progressively decline during the extended cycling of CLL cells, we employed CFSE-based division subset analysis (32, 34, 46, 57). A representative experiment with ODN + IL15-stimulated M-1328 (Figure 1C) shows that BCL2 expression notably dropped with progressive divisions (RMFI = 39 for the activated cells with no division versus RMFI = 10 for subpopulations with $\geq$5 divisions). MCL1 protein was either sustained or slightly increased upon cycling (RMFI = 3.7 for the undivided subset versus RMFI = 4.1 in cells with >5 divisions). To assess the consistency of this ODN + IL15-driven decline in BCL2 protein during cycling, a total of 14 different CLL clonal populations were examined (Figure 1D). For most of these same clonal populations (12 of 14), a parallel analysis of MCL1 expression was made (Figure 1E). Plots of individual CLL clonal responses reveal that the maximal BCL2 expression is universally observed in the undivided subset, with all populations exhibiting a division-linked decline in BCL2 levels (Figure 1D). In contrast, MCL1 expression was sustained and sometimes slightly elevated during cycling (Figure 1E).

Parenthetically, it warrants noting that the majority of viability-gated, undivided cells in these activated cultures are not quiescent; rather, they characteristically manifest some size enlargement, as measured by increased forward light scatter upon flow cytometry (or visibly increased size by phase microscopy) over CLL cells cultured in medium or IL15 alone (data not shown; [2,34,35]). Quiescent normal lymphocytes characteristically manifest increased cell volume as they enter the G1 phase of the cell cycle [37–39], suggesting that the undivided fraction within these stimulated CLL cultures may have entered G1 but have been thwarted from cycling due to cell cycle blocks at the G1/S or G2/M transitions.

Figure 1F summarizes pooled findings from BCL2 assessments with diverse cycling CLL populations, with accompanying statistical analyses. Of note, statistically significant differences ($p \leq 0.02$) were noted both when BCL2 levels within undivided cells were compared to levels in each ensuing division as well as when progeny with sequential divisions were compared. A similar analysis of MCL1 expression (Figure 1G) revealed no statistically significant change linked to division.

Because in patients, IGHV unmutated (U-CLL) and mutated (M-CLL) clones typically differ in clinical aggressiveness (U-CLL > M-CLL) [40–42], we subdivided the full CLL cohort into U-CLL and M-CLL subtypes and compared the latter for differences in BCL2/MCL1 protein expression during cycling. Summarized data (Figure 1F) show that BCL2 is comparably expressed in U-CLL and M-CLL, with both groups manifesting similar division-linked reductions. However, a difference emerges between these IGHV subsets concerning MCL1 expression (Figure 1G). This pro-survival protein was, on average, higher in U-CLL cells and more apt to rise during cycling; however, only in cells representing two divisions did this U-CLL versus M-CLL difference reach borderline statistical significance ($p$ = 0.09 by one-sided, unpaired *t*-test). Thus, while IGHV mutation status has no impact on the division-linked decline in BCL2, it may influence MCL1 expression during cycling, with U-CLL > M-CLL.

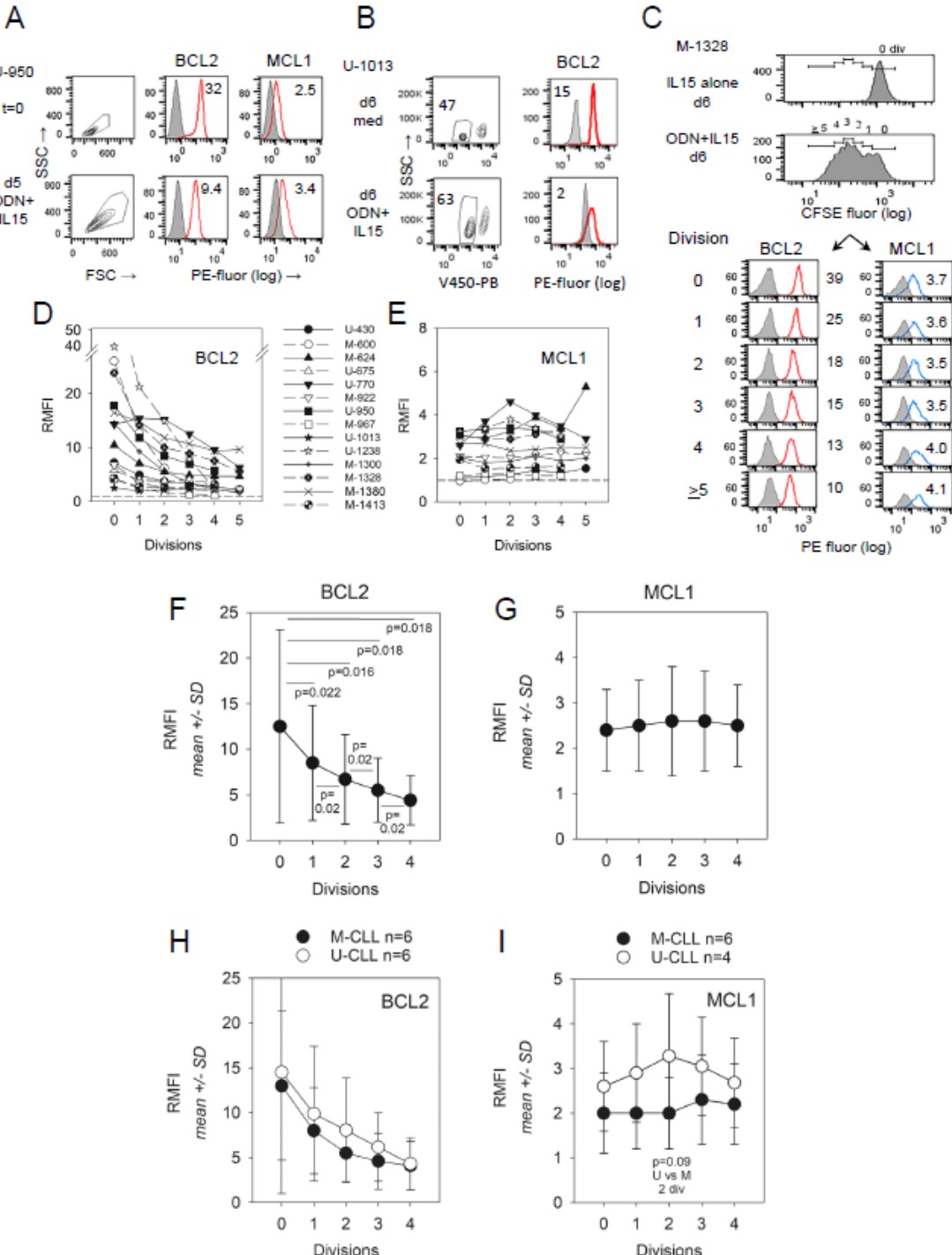

**Figure 1.** Distinct modulation of BCL2 and MCL1 protein levels during ODN + IL15-induced CLL cell cycling. (**A**) U-CLL950 cells were evaluated for intracellular BCL2 and MCL1 protein both at d0 and following 5 d of ODN + IL15 stimulation. PE-fluorescent intensity representing specific mAb (open red histograms) or respective Ig isotype controls (grey filled) was measured on viability-gated cells by flow cytometry. Inserted values represent the ratio median fluorescence intensity (RMFI) of specific versus control mAb. (**B**) Comparison of BCL2 levels in U-CLL1013 cells following d6 culture with medium alone or ODN + IL15 (non-activating and activating conditions), respectively [2,34,35]. (**C**) Two-color flow cytometry with CFSE-labeled M-CLL1328 CLL cells was used to monitor BCL2 and MCL1 protein levels as a function of division status. Top 2 histograms display CFSE fluorescence within viability-gated cells from d6 cultures with IL15 alone or activating

ODN + IL15. While the former remains undivided, the latter shows over 5 cycles of division, as noted earlier with this clonal population [2]. Bottom 6 histograms represent BCL2-PE (left column) and MCL1-PE (right column) fluorescence (shaded = Ig control) within gated division subsets of ODN + IL15-stimulated cells (values = RMFI). (**D,E**) Summary plots of RMFI values per division for all ODN + IL15-stimulated CLL populations tested ((**D**) = BCL2; (**E**) = MCL1). (Of note: Maximal BCL2 protein level appeared to be an intrinsic attribute of the individual CLL clone since RMFI values were quite similar when 5/6 CLL populations were assessed for BCL2 in two separate experiments. For the one exception (M-CLL1328), where a 3-fold difference in RMFI values was noted between experiments, a mean value was used. (**F,G**) Plots showing mean RMFI ($\pm$SD) for (**F**) BCL2 and (**G**) MCL1 expression per each division subset of the CLL panel represented in (**D,E**). Paired, 2-tailed *t*-tests were used for assessing the statistical significance of differences between levels in the undivided subset as opposed to each division subset, as well as BCL2 expression changes within cells at successive divisions (1 vs. 2 div; 2 vs. 3; and 3 vs. 4 div); all differences were found to be statistically significant. In (**F–I**) below, 2 CLL populations (M-CLL600 and M-CLL1300), whose individual BCL2/MCL1 values are plotted in D-E, were not used for statistical analyses of CLL pools in (**F–I**): M-CLL600 due to insufficient cell numbers for assessment at $\geq$3 divisions; M-CLL1300 due to unreliability of assessing cells with $\geq$4 divisions (>50% CD19-negative; Supplementary Figure S3). (**H**) Comparative BCL2 levels in pooled M-CLL (n = 6) and pooled U-CLL (n = 6), gated by division subsets; no statistically significant difference was noted by unpaired *t*-test (2-sided or 1-sided). (**I**) Comparative MCL1 levels in pooled M-CLL (n = 4) and U-CLL (n = 6). While mean MCL1 expression in all U-CLL division subsets exceeded that of M-CLL, the difference was only of borderline statistical significance at 2 divisions (*p* = 0.09 by unpaired, 1-sided *t*-test). In a separate statistical analysis in which M-600 and M-1300 were included in pools for comparing MCL1 levels in U-CLL versus M-CLL cohorts, the difference between U-CLL and M-CLL at 2 divisions became statistically significant (*p* = 0.04) by unpaired, 1-sided *t*-test.

## 2.1. Sensitivity of non-Cycling and Cycling CLL Cells to Specific Inhibitors of BCL2 or Surviving

To test this study's additional hypothesis that declining BCL2 levels during active clonal expansion affects susceptibility to venetoclax-induced apoptosis, we assessed the percent viability and absolute viable cell recovery [2] within d5(d6) CLL populations pulsed with either venetoclax or vehicle alone 24–48 h prior to culture harvest (Figure 2). Cells harvested from cultures with non-stimulatory IL15 alone and proliferation-inducing synergistic ODN + IL15 were compared for sensitivity to the BCL2 inhibitor. Likewise, cultures were examined for sensitivity to YM155, a specific inhibitor of survivin synthesis [43], since the expression profile of survivin within body cells suggests it might be a more suitable therapeutic target than MCL1. Although MCL1 appears to confer CLL resistance to venetoclax [27–29], MCL1 is also essential for the maintenance of cardiac cells [44,45] and long-lived plasma cells within BM [46,47], making its targeting problematic. In contrast, survivin is absent in non-proliferating body cells and highly expressed in malignancies [48], including CLL pseudofollicles [21]. Past observations that CD40 signaling induces survivin in quiescent CLL cells [21] suggested that signaling from proximal T cells was required for its expression in this leukemia [21,43]. Nonetheless, alternative means of CLL activation, e.g., ODN + IL15, can upregulate or sustain its baseline expression within quiescent CLL cells exposed to medium or IL15 alone (Supplementary Figure S4).

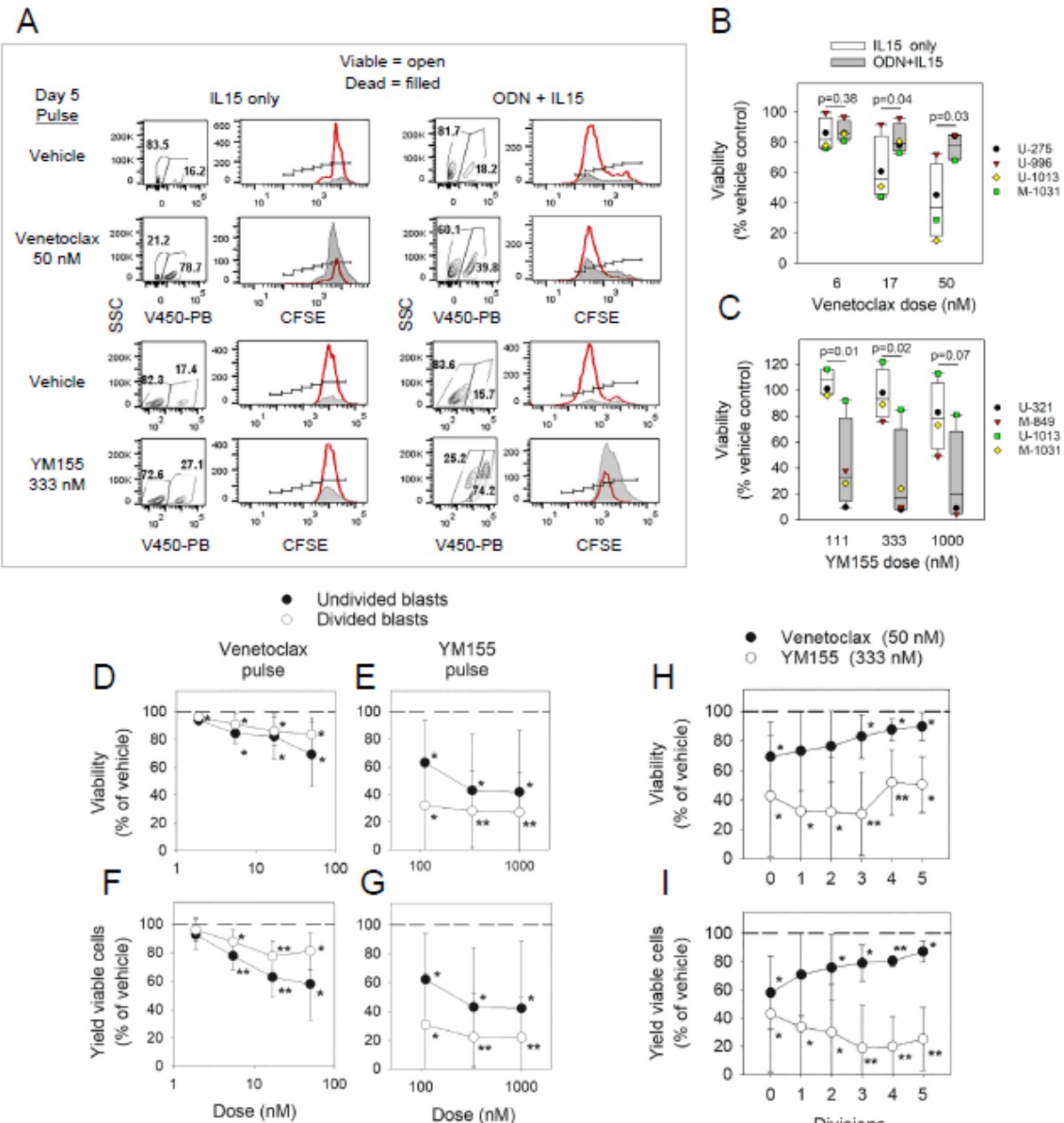

**Figure 2.** BCL2 inhibitor and survivin inhibitor preferentially target non-cycling and cycling CLL cells, respectively. (**A**) M-CLL1031 experiments comparing quiescent (IL15 only) and cycling (ODN + IL15) d5–6 cultures for sensitivity to venetoclax (50 nM) and YM155 (333 nM): inhibitors added during the last 24–36 h of d5(6) cultures. Following harvest, CFSE-labeled cells were gated into viable and dead subsets by SSC and V450-Pacific Blue dye exclusion, and both subsets were assessed for division (open histograms = viable; shaded histograms = dead). (**B,C**) Pooled viability analysis of CLL cultures stimulated with either IL15 or ODN + IL15 and subsequently pulsed with varying doses of either (**B**) venetoclax (n = 3 CLL exp: U-CLL966, M-CLL275 and M-CLL1031) or (**C**) YM155 (n = 3 CLL exp: U-CLL321, M-CLL849, and M-CLL1031). Data expressed as % of control with vehicle (mean ± SD). In these pooled CLL experiments, inhibitors were added during the last 36–48 h of d5–6 cultures. (Note: in a separate experiment with U-CLL1013 cells cultured for 39 h with medium or IL15 alone, both populations were equally sensitive to venetoclax; data not shown). * indicates statistically significant difference in viability when compared to control cultures pulsed with vehicle (in each exp normalized to 100%). pooled analysis of gated undivided and divided blasts from ODN

+ IL15-stimulated cultures for (**D**,**E**) viability and (**F**,**G**) absolute yield of viable blasts following pulse with venetoclax (**D**,**F**) or YM155 (**E**,**G**). * indicates $p < 0.05$ and $> 0.001$ while ** indicates $p < 0.001$ when values in inhibitor-pulsed cultures are compared with parallel cultures with vehicle alone. (**H**,**I**) ODN + IL15-induced blasts with differing division histories were compared for (**H**) overall viability and (**I**) absolute yield of viable cells per culture following exposure to venetoclax or YM155. Data expressed as % of values in parallel cultures pulsed with vehicle alone (mean $\pm$ SD). Experiments with venetoclax (n = 5) involved the following CLL: U-284, U-675, U-966, U-1013, and M-1031. Experiments with YM155 (n = 5) involved U-321, M-849, U-1013, M-1031, and U-1692.

Figure 2A displays both viability plots and CFSE fluorescence histograms of gated viable and dead cells from d5 cultures of M-CLL1031 cells treated with venetoclax, YM155, or vehicle alone 24 h prior to cell harvest. Of note, viability in cultures exposed to IL15 alone is quite notably reduced by exposure to venetoclax but only slightly impaired by YM155 when compared to parallel control cultures pulsed with vehicle (Figure 2A, two left columns). In contrast, in ODN + IL15-stimulated cultures undergoing significant cycling (Figure 2A, two right columns), venetoclax only slightly impairs survival, while the survivin inhibitor YM155 quite effectively does so. Moreover, an inspection of CFSE division profiles of ODN + IL15 cultures, pulsed with vehicle or YM155 (Figure 2A right, row 3 vs. 4), reveals that YM155 aborts continued cell cycling in remaining viable progeny. Dose–response analyses involving several CLL populations (Figure 2B,C) confirm that the efficacy of these inhibitors at compromising CLL viability is strongly influenced by whether CLL cells received stimuli for growth: venetoclax is less effective on ODN + IL15-activated CLL cells, while conversely, YM155 shows greater efficacy. Thus, the relative effectiveness of these inhibitors is strongly influenced by whether CLL cells have received robust activation signals or not.

To discern whether undivided and divided cells within all activated CLL populations will differ in sensitivity to venetoclax and YM155, activated cultures of diverse CLL clonal populations were tested with a range of venetoclax and YM155 doses (Figure 2D–G). Statistical analysis of the pooled findings shows that each inhibitor significantly reduces viability in the undivided as well as divided fractions of activated CLL cell cultures, either when assessments are made of (a) percent viability in the cells (viable + dead), gated to remove cell debris (Figure 2D,E) or (b) absolute viable cell recovery within each culture (Figure 2F,G). These studies indicate that YM155 is notably more effective at compromising each of the latter in the divided cell fraction than in undivided cells, while the converse is so with venetoclax. It warrants noting that while venetoclax (50 nM) compromises percent viability in activated but undivided cells by ~15% (Figure 2D), this is less than its apoptosis-inducing effects within quiescent cultures with IL15 alone (~ 60% decline in viability) (Figure 2B). This finding is consistent with other reports that survival proteins other than BCL2 are elevated within 1 to 3 days after CLL cell activation [26–29].

Because BCL2 protein levels progressively decline during CLL cycling (Figure 1D,F), we examined whether activated CLL clonal populations manifest progressively lesser vulnerability to venetoclax as they continue dividing. Data presented in Figure 2H,I indicate that venetoclax sensitivity indeed declines with greater division history. This contrasts with data from the percent viability analysis of YM155-treated cultures (Figure 2H), which suggests that apoptotic effects of YM155 are relatively independent of division frequency. Notwithstanding, when absolute viable cell recovery is assessed (Figure 2I), YM155 preferentially compromises the yield of highly divided blasts, a finding consistent with the blocked cycling of remaining viable cells, observed in Figure 2A.

Phase microscopy images of day 5 ODN + IL15-stimulated CLL cultures, receiving a 24 pulse with YM155 (333 nM) or vehicle alone prior to observation (Figure 3) reveal that exposure to YM155 elicits shrunken cells with fragmented nuclei and membrane blebs, consistent with apoptosis (Figure 3B). Also, present in what appear to be larger viable cells are intracellular structures resembling statically aligned chromosomes (black stars; Figure 3B), not seen in parallel cultures pulsed with vehicle (Figure 3A). These might represent cells with a YM155-induced block in the G2 $\rightarrow$ M transition [49].

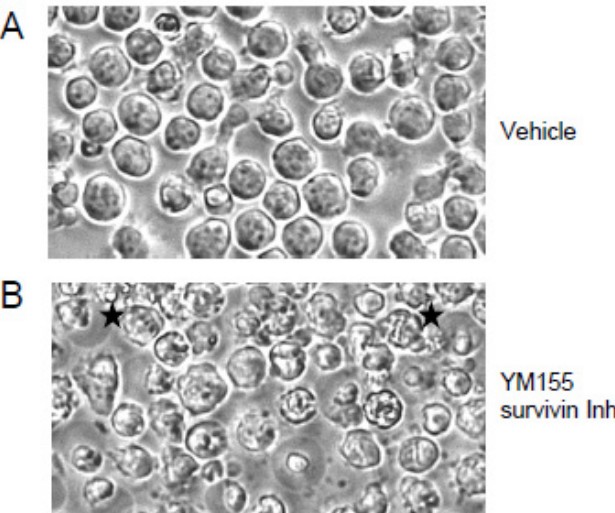

**Figure 3.** Phase microscopy of ODN + IL15-stimulated cultures exposed to YM155 survivin inhibitor or vehicle alone. U-CLL1692 cells were stimulated with ODN + IL15; pulsed on d4 with vehicle alone (DMSO) (**A**) or YM155 (333 nM) (**B**); and examined by phase microscopy 24 h later. Note the evidence of shrunken cells with membrane blebs (apoptosis) in YM155-treated cultures. Also evident are cells with intracellular structures reminiscent of statically aligned chromosomes during the M phase of the cell cycle (black stars).

### 2.2. Impact of CLL del(13q) on BCL2 and MCL1 Protein Levels during ODN + IL15-Driven Cell Clonal Expansion

As a step toward unraveling mechanism(s) for reduced BCL2 protein during cycling, we tested for *miR15a/miR16-1* involvement in the decline by segregating assessed CLL populations into cohorts based on the presence of del(13q), which removes coding regions for *miR15a/miR16-1* [6–9] (Figure 4). Important molecular studies indicated that these miRs repress BCL2 (and MCL1) expression in CLL [50].

Histogram plots of individual ODN + IL15-stimulated CLL populations (Figure 4A,B) and box-plot analysis of pooled data (Figure 4C) show that BCL2 protein is, on average, more highly expressed in stimulated del(13q)+ CLL than in del(13q)−/− CLL; a finding consistent with past observations, involving freshly isolated CLL cells, that del(13q) influences BCL2 protein levels (del(13q)+ > del(13q)−/−) [51]. A comparison of del(13q) influence on the gated undivided and divided cell subsets (Figure 4C) shows that both these fractions are affected, with a borderline level of statistical significance (Figure 4C). Nonetheless, no statistically significant difference ($p = 0.84$) was noted between del(13q)+ and del(13q)−/− CLL cohorts in the magnitude of the division-linked decline (latter calculated by comparing BCL2 levels in undivided cells versus those representing three divisions) (Figure 4D), suggesting that additional factors are also contributing to diminished BCL2 expression during cycling.

Because MCL1 is also a reported target of *miR15* and *miR16* [8,52], its protein levels were also compared within activated del(13q)+ and del(13q)−/− CLL populations (Figure 4E,H). While a higher proportion of cycling del(13q)$^+$ CLL cells (6/8 = 75%) than del(13q)−/− cells (2/4 = 50%) displayed elevated MCL1 (RMFI ≥ 2), subsequent box plot and statistical analysis showed no statistically significant difference between cohorts with/without del(13q), either when maximal MCL1 attained (Figure 4G) or division subset with maximal MCL1 (Figure 4H) were assessed as parameters. Unlike long-lived BCL2 ($t_{1/2} = 20$ h) [53], MCL1 is a very short-lived protein ($t_{1/2} = 1$ h) [54], known to be highly regulated by several post-translational changes affecting its stability [55]. Possibly, these obscure any influence of Chr13-encoded *miR15a/miR16-1* on expressed MCL1 protein.

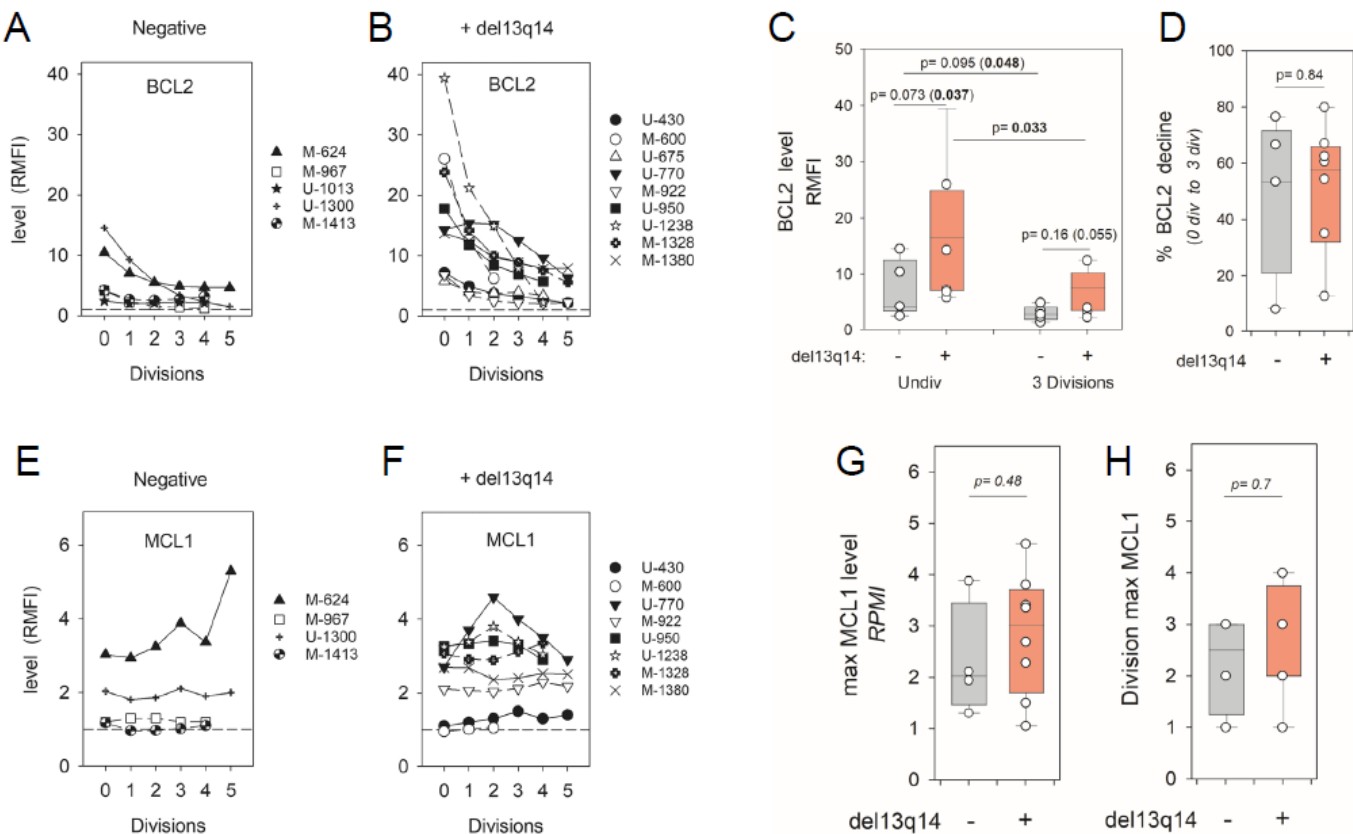

**Figure 4.** Deletion of Chr13 loci encoding *miR15/miR16* is linked to greater BCL2 protein during clonal expansion. (**A**,**B**) BCL2 expression (RMFI) versus division frequency within ODN + IL15-stimulated CLL populations segregated into (**A**) del(13q)−/− and (**B**) del(13q)+ CLL cohorts (**C**) Box plot analysis of pooled experiments comparing BCL2 within undivided and divided blasts of del(13q)+ and del(13q)−/− groups. Note: BCL2 expression in cells with 3 divisions was used as representative of dividing cells because (a) CLL populations vary in the extent of cycling and (b) all stimulated CLL (excepting M-CLL600 and U-CLL1300) had reliable numbers of cells representing 3 divisions for analysis. Statistical comparisons of BCL2 levels within undivided versus divided cells of the same cohort involved a 2-sided, paired *t*-test. When differing cohorts were compared, a 2-sided, unpaired *t*-test was employed to determine statistical significance. (*p* values indicated in parenthesis represent the use of a 1-sided *t*-test), (**D**) Values for % decline in BCL2 obtained by comparing RMFI in cells with 3 divisions versus RMFI in respective undivided CLL cells. (**E**,**F**) MCL1 protein expression versus division frequency in (**E**) del(13q)−/− and (**F**) del(13q)+ CLL. (**G**) Box plot analysis of maximal MCL1 within viable cells of del(13q)+ versus del(13q)−/− CLL cohorts stimulated with ODN + IL15. (**H**) Comparison of del(13q)+ and del(13q)−/− CLL cohorts for cell division at which maximal MCL1 was noted.

An examination of known chromosomal anomalies of the CLL populations under study (Table 1) reveals other factors complicating our assessment of how *miR15a/miR16-1* loss affects BCL2 (and MCL1) expression. First, del(13q) is typically present in only 1/2 chromosomes (heterozygous expression), and del(13q)+ CLL populations vary in the fraction of the malignant clone exhibiting the deletion. Second, CLL clonal populations differ in expression of other chromosomal anomalies, e.g., Trisomy 12 and del11q22, that might contribute to variations. Additionally, although del(17p) was uniformly absent from our study cohorts, comprehensive mutation analysis was not made of mutations prevalent in CLL, e.g., within genes for p53(TP53), ATM, NOTCH1, SF3B1, and BIRC3 [56,57].

**Table 1.** Known genetic characteristics of human CLL populations under study.

| CLL [A] | IGHV Status | IGHV Gene | del13q14 [B] | Tri12+ [B,C] | del11q22 [B] | del17p [B] | ATM [D] Mut | Therapy (Mo Prior) |
|---|---|---|---|---|---|---|---|---|
| 275 | M | 3-30 | +(16% ho; 6% het) | neg | neg | neg | nd | |
| 321 | U | 4-34 | neg | +(53%) | neg | neg | nd | |
| 430 | U | 1-69*01 | +(97% het) | neg | +(10% het) | neg | + | 276 mo |
| 515 | U | 4-39*01 | +(6.5%) | nd | nd | nd | nd | |
| 600 | M | 4-34*01 | +(99% ho) | neg | neg | neg | nd | |
| 624 | M | 3-7*01 | neg | +(10%) | neg | neg | WT | |
| 625 | U | 1-69 | neg | neg | +(80% het) | neg | nd | |
| 631 | U | 3-20 | neg | neg | +(90% het) | neg | nd | |
| 675 | U | 3-23*01 | +(80% het) | neg | +(19% het) | neg | nd | 39 mo |
| 770 | U | 3-15*01 | +(89% het) | neg | neg | neg | + | |
| 849 | M | 3-72 | +(68% het) | +(96%) | neg | neg | nd | 72 mo |
| 887 | U | 3-30-3*01 | +(30% het) | neg | +(27%) | neg | WT | |
| 922 | M | 4-34*07 | +(45% het) | neg | neg | neg | nd | |
| 950 | U | 2-5*10 | +(89% het) | neg | neg | net | nd | 8 mo |
| 967 | M | 3-7*01 | neg | neg | neg | neg | nd | |
| 996 | U | 1-69 | neg | neg | neg | neg | nd | |
| 1013 | U | 3-33 | neg | +(52%) | neg | neg | nd | |
| 1031 | M | 4-39 | neg | neg | neg | neg | nd | |
| 1158 | U | 3-15*01 | +(97%) | nd | +(48%) | nd | nd | |
| 1238 | U | 3-30-3*01 | +(55% het) | neg | +(69% het) | neg | nd | |
| 1239 | U | 3-30*03 | neg | neg | neg | neg | WT | |
| 1300 | M | 3-7*02 | neg | neg | neg | neg | WT | |
| 1328 | M | 4-61*01 | +(86%) | neg | neg | neg | nd | |
| 1380 | M | 3-7*01 | +(50% ho; 22% het) | neg | neg | neg | nd | |
| 1413 | M | 4-39*01 | neg | +(20%) | neg | neg | nd | |
| 1529 | M | 4-59*01 | +(30% ho; 60% het) | neg | neg | neg | nd | |
| 1692 | U | 2-7 | neg | +(85%) | neg | neg | nd | |
| 1993 | U | 3-11*01 | +(90%) | nd | nd | +(90%) | nd | |
| 2018 | M | 3-84*01 | +(61%) | +(61%) | nd | nd | nd | |

[A] "CLL clone" represents a CD19+/CD5+ population expressing a uniform IGHV sequence; it does not exclude the presence of subclones. CLL cells were obtained from patients prior to therapy, except those indicated. [B] Chromosomal anomalies determined by prior FISH analyses. [C] ATM; the gene for ataxia telangiectasia mutated: CLL430 mutation in exon 64; CLL770 mutations in exon 41 and 50 [2]. [D] In CLL with sequenced *ATM*, + indicates presence of mutations and WT indicates wild-type sequence.

### 2.3. ODN + IL15-Stimulated CLL Cells Express Elevated Protein Levels of p53 Transcription Factor (TF)

Molecular studies by others, using doxorubicin-treated B cell lines and primary CLL cells, demonstrated that p53 directly transactivates *miR15/miR16,* resulting in BCL2 mRNA/protein [58,59]. This led to the hypothesis that activation-driven increases in p53 protein within CLL cells, such as found in vivo within germinal centers [60] and in vitro within human B cell cultures activated by division-eliciting T cell independent stimuli [32], might drive BCL2 repression during CLL cycling.

To test the above premise, we measured p53 protein within ODN + IL15-stimulated CLL cultures, both by intracellular staining for nuclear p53 and by immunoblotting cell lysates for p53α (MW ~ 52kDa) (Figure 5). Flow cytometric studies showed that while p53 protein was minimally evident in quiescent cells prior to activation (Figure 5E) or from cultures with IL15 alone (non-stimulatory conditions) (Figure 5A–F), p53 protein was significantly heightened in ODN + IL15-stimulated cultures (Figure 5A–F). Two-color assessments of p53 protein as a function of division cycles (Figure 5B–D) reveal that p53 rises within both undivided and cycling cells of ODN + IL15-activated cultures, with the highest levels typically associated with cycling. A time course experiment with U-CLL430 (Figure 5C,D) reveals that p53 protein is upregulated by at least day 3 within the undivided cells of ODN + IL15-stimulated cultures, and its levels increase further with division. This contrasts with lower p53 protein levels in CLL cells cultured with either IL15 alone or ODN

alone (Figure 5C–E). Box plots comparing p53α/actin ratios within lysates of ODN + IL15 activated CLL cell cultures versus lysates of quiescent CLL cell cultures (Figure 5F) illustrate the consistent pattern of elevated p53α protein in activated cultures (borderline statistical significance; *p* = 0.06).

In striking contrast to the elevated p53 protein found in ODN + IL15-activated CLL cells, qPCR assessments of specific mRNA showed CLL cell activation by ODN + IL15 was linked to significantly reduced TP53 mRNA, both when comparisons were made to freshly isolated CLL (Figure 5G,H, left) or parallel ODN-stimulated CLL cultures without IL15 (Figure 5G,H, right). (Reduced *TP53* mRNA was also noted when d3 cultures with ODN + IL15 were compared to parallel cultures with ODN alone; data not shown.) The opposing findings regarding *TP53* mRNA and p53 protein expression strongly suggest that the protein's augmentation during activation represents its greater stabilization, e.g., by ATM activation following the oxidative stress and DNA damage accompanying robust B cell activation [32,61]. Consistent with this, levels of phosphorylated (activated) ATM and stabilized p53 (p-Ser15-p53) rise significantly prior to division, both in activated cultures of normal human B cells [32,62] and CLL cells stimulated by CD40L + IL15 [63]. A potential mechanism for this reduced *TP53 mRNA* levels is miR15/miR16-mediated negative feedback suppression of p53 (*TP53*) [58]; this will be addressed later.

### 2.4. Early Effects of IL15 and Downstream STAT5 and PI-3K on BCL2/MCL1 mRNA Levels

When taken together, the above observations (Figure 5A–F) that levels of nuclear p53 TF protein rise in an IL15-dependent manner (both prior to division and perhaps particularly within cycling cells) in ODN + IL15 stimulated CLL cultures and the earlier molecular finding that p53 TF directly transactivates *miR15/miR16* in CLL cells [58] suggest that an IL15-driven p53 → *miR15/miR16* pathway contributes to declining BCL2 protein as CLL cells undergo cycling.

Consequently, we sought evidence for diminished *BCL2* mRNA early after IL15 signaling. To facilitate this, unstimulated CLL cells were primed for 20 h with ODN. We had established earlier [35] that this priming interval is required for CLL cell upregulation of two IL15 signaling receptors, CD122 (shared by IL2 and IL15) and CD215 (IL15-specific high-affinity IL15Rα), above their negligible levels in quiescent CLL cells [35]. Following 20 h ODN priming, CLL cells were pulsed for either 4 h or 20/28 h intervals (or only medium). Subsequently, *BCL2* (and *MCL1*) mRNA levels were assessed by quantitative RT-PCR (qPCR) (Figure 6). Both when data are expressed as raw Δ Ct values (Figure 6A) or as more easily interpreted fold-change values (Figure 6B), a statistically significant decline in BCL2 mRNA was evidenced following a 20/28 h pulse with IL15 (*p* = 0.006 and *p* = 0.002, respectively). A lesser, but still statistically significant, decline in *MCL1* mRNA was also noted following the 20/28 h IL15 pulse by fold change (*p* = 0.03) (Figure 6B).

To discern whether downstream IL15 signaling mediators, STAT5 and/or PI-3K [34], were involved in this IL15-driven decline in *BCL2* mRNA, pharmacologic inhibitors were used. Each of two STAT5 inhibitors attenuated IL15-triggered repression of *BCL2* (Figure 5C; *p*= 0.002 for pimozide; *p*= 0.06 for weaker Stat5 Inh II [34]) as did PI-3K inhibitor, LY294002 (*p* = 0.03). Together, these findings suggest that BCL2 protein repression within ODN + IL15-activated CLL cells (Figure 1) is mediated, at least in part, by an IL15→ STAT5/PI-3K pathway that is also critical for extended growth [34,35].

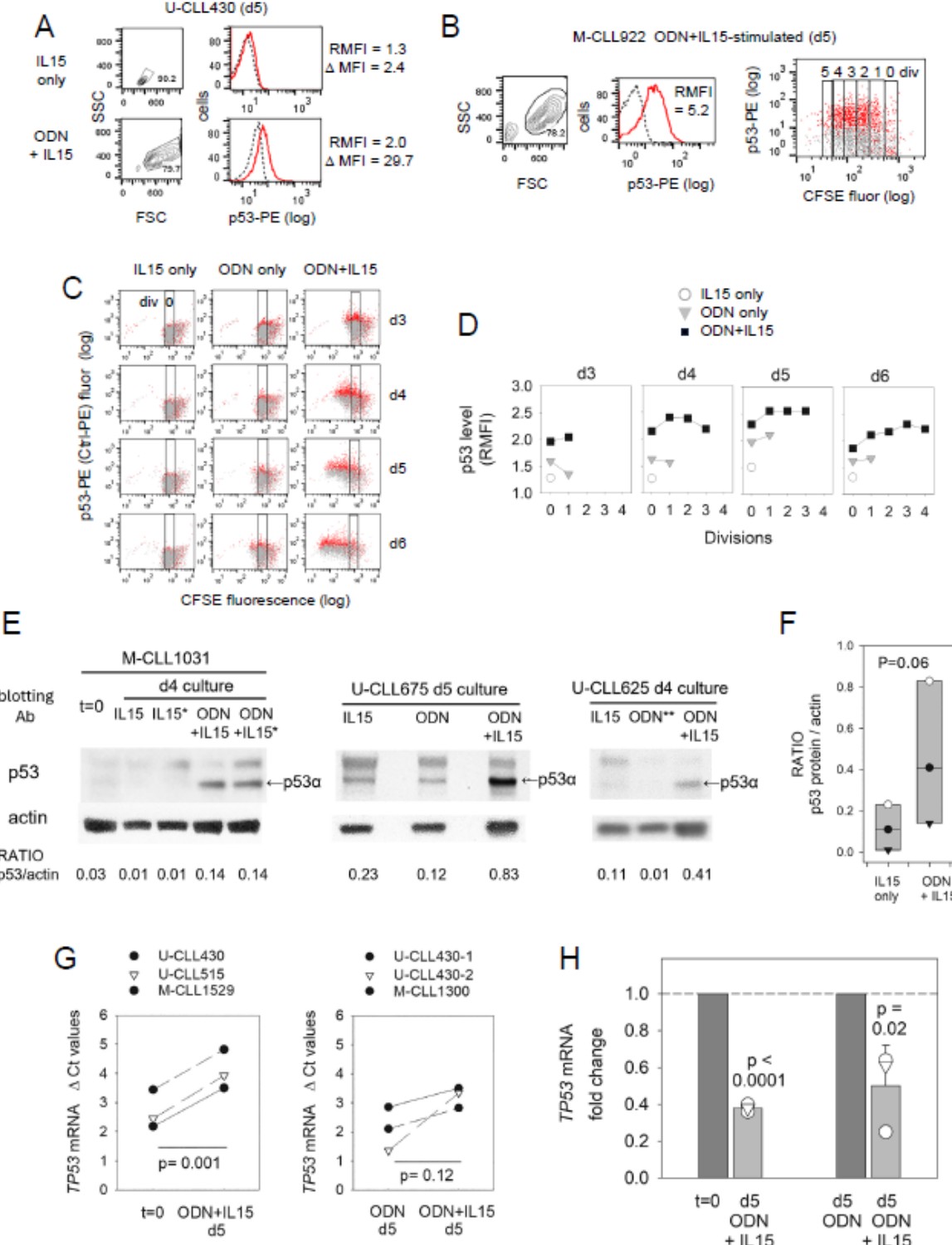

**Figure 5.** p53 protein rises during ODN + IL15-induced B-CLL cycling, while *TP53* mRNA declines. (**A**) p53-protein PE fluorescence (solid lines) versus control IgG (dashed) in viability-gated U-CLL430 cells from d5 cultures with IL-15 alone or ODN + IL15. Values represent RMFI and Δ MFI (difference between geometric MFI with specific versus control mAb). (**B**) p53 protein within CFSE-labeled M-CLL922 cells from 5 d cultures with ODN + IL15. Two-color dot plot (right) reveals p53 protein (red) within viability-gated cells of gated division subsets (control stain = grey). (**C**) Kinetic analysis of p53 protein expression within CFSE-labeled U-CLL430 cells cultured for d3 to d5 days with IL15 alone, ODN alone, or ODN + IL15. Rectangle designates the undivided subset. In these dot plots,

staining with p53-PE (red) overlaid with Ig Ctrl-PE (grey) staining in cells of varying divisions. (**D**) Summarized display of data in (**C**) as p53 levels (RMFI) per division over d3 to d6. (Of note, excepting d3, all daily p53 staining analyses included a defrosted sample of the CL-01 B cell line, used for standardization of p53 staining; all values for adjustment were < 2-fold.) (**E**) Levels of p53α (canonical isoform) were measured by densitometric analysis of both p53 and β-actin blots followed by calculation of p53α/actin ratio for each assessed lysate (M-CLL1031 exp with 31 μg lysate/lane where * indicates cultures supplemented with caspase-2 inhibitor; U-CLL675 exp with 20 μg/lane; U-CLL625 exp with 15 μg/lane, excepting ODN ** only, with 6 μg/lane). The bands above p53α likely represent its ubiquitinated forms [64], while any bands below p53 isoforms from differential splicing [65]. (**F**) Box plot summarizing data (RATIO of p53/actin) from blotting experiments comparing lysates from IL15-only cultures to those from parallel ODN + IL15 cultures. Box plots are overlaid with values from individual lysates. Statistical comparisons of IL15 only versus IL15 + ODN involved the non-parametric Mann–Whitney Rank Sum test. (**G,H**) q-PCR assessments of *TP53* mRNA in CLL cells; (**G**) ΔCt values computed from the difference between threshold cycle for test *TP53* cDNA and threshold cycle for reference β-actin (*ACTB*); (**H**) *TP53* mRNA fold change calculated using within $2^{-\Delta\Delta Ct}$ method [66], to facilitate comparison of *TP53* mRNA levels in cells with versus without IL15 exposure. Statistical analysis by paired, 2-sided *t*-test.

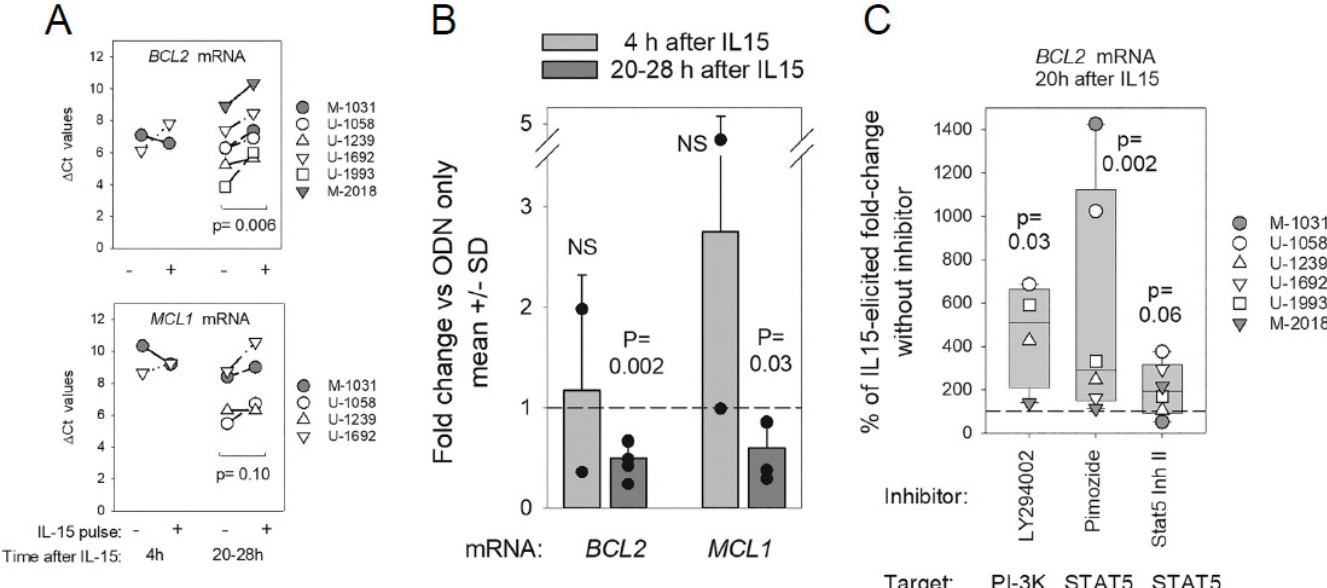

**Figure 6.** IL-15-triggered STAT5/PI-3K pathway reduces BCL2 mRNA within ODN (TLR9)-primed CLL cells. These experiments sought evidence for altered BCL2 mRNA levels after IL15 exposure to ODN-stimulated CLL cells. ODN priming was important because we earlier reported that 20 h ODN exposure elicited the NF-kB-dependent upregulation of IL15 receptors (CD122 and IL15Rα) above negligible baseline levels on resting CLL cells [35]. Any evidence that short-term IL15 exposure resulted in dampened BCL2 mRNA might be explained by an IL15-fostered rise in *BCL2* repressive miR (e.g., miR15a/miR16-1) [58]. Our rationale for suspecting a critical role for IL15 cytokine in mediating BCL2 repression was based upon the following. First, IL15 signaling was necessary for a significant rise in p53 TF protein within CLL cells receiving ODN signals (Figure 5C–E), as well as for significant CLL clonal expansion within ODN-stimulated cultures [2,34,35]. Second, the IL15 boost in p53 TF (a direct transactivator of miR15a/miR16-1 synthesis) was detected in undivided CLL cells by at least day 3, even prior to IL15-driven divisions characterized by sustained/further elevated p53 TF protein (Figure 5C,D). For these experiments, quiescent CLL cells were primed with ODN for 20 h and subsequently pulsed with IL-15 (or medium alone) for 4 h or 20–28 h intervals before total RNA was harvested and BCL2 and MCL1 mRNA quantified by specific q-PCR. (**A,B**) ΔCt values for

(**A**) BCL2 mRNA and (**B**) MCL1 mRNA within IL15-pulsed or un-pulsed cultures at differing intervals after the IL15 pulse. A paired, 2-sided t-test was used to compare ΔCt values from primed cultures with/without IL15. (**C**) More intuitive fold-change values, calculated with the $2^{-\Delta\Delta Ct}$ method [66], better reveal the altered levels of BCL2 and MCL1 mRNA at 20–28 h after the IL15 pulse. Bars represent the mean ± SD of the diverse experiments, with overlaid symbols representing values from individual CLL. Statistical significance was determined by the non-parametric Mann–Whitney rank sum test. (**C**) Experiments with STAT5/PI-3K inhibitors examined whether IL15 activation of STAT5/PI-3K pathways is critical for IL15-triggered BCL2 mRNA repression, as earlier noted for IL15-facilitated growth of ODN-primed CLL cells [34]. Specific inhibitors of PI-3K (LY294002), STAT5 (pimozide or STAT5 INH II), or vehicle alone were added to ODN-primed cultures 30 min prior to a 20 h pulse with IL-15. By q-PCR, the yield of *BCL2* mRNA in IL15 pulsed cultures was compared to the yield in ODN-primed cultures that had been exposed to DMSO but not IL15. Consequently, IL15-induced fold change in *BCL2* mRNA was determined. (Note: in cultures pulsed with DMSO alone, IL15-elicited fold change in BCL2 mRNA was 0.57 ± 0.04 (mean ± SD) for the diverse experiments). When pooling inhibitor results from these experiments, data from each CLL experiment was normalized by comparing IL15-induced fold change "with inhibitor" to IL15-induced fold change "with DMSO alone". From the latter determinations, plotted values for % of IL15-elicited fold change without inhibitor" could be calculated. Bar blots display median/range values for the above determination noted with the diverse CLL evaluated, with overlaid symbols representing values for individual CLL evaluated. The dotted horizontal line represents the fact that, with this approach, all values for "IL15-induced fold change without inhibitor" are effectively normalized to 100%. Any rise in this relative percentage indicates that the inhibitor blocked the IL15-induced *BCL2* mRNA decline. *p* values for statistical significance were determined using a non-parametric Mann–Whitney rank sum test.

### 2.5. Evidence for miR15a/miR16-1 Mediated Feedback Repression of p53 TF

Of interest and consistent with past molecular evidence that *miR15/miR16* mediates negative feedback control on *TP53* synthesis [58], we discovered significant differences in p53 protein levels within del(13q)+ and del(13q)−/− CLL populations. Whereas most del(13q)+ CLL (7/8) displayed prominent division-linked increases in p53 (Figure 7B), which were highly significant (divided > undivided; *p* = 0.004) (Figure 7C), this was not observed for del(13q)−/− CLL populations (Figure 7A–C) (divided vs. undivided; *p* = 0.31). Because 2/5 CLL clonal populations within the del(13q)−/− cohort exhibited both aberrantly high p53 protein and Trisomy12+ (Tri12+) (Figure 7D versus Figure 7E) (factors that in later discussion we consider are linked), these two Tri12+ clones were excluded from a reassessment of del(13q) influence on p53 protein (Figure 7H). Despite this step, p53 levels within the del(13q)−/− cohort only negligibly increased with division (divided > undivided; *p* = 0.16). However, a statistically significant difference in p53 protein levels within divided cells of del(13q)+ versus del(13q)−/− cohorts emerged (del13+ > del13−/− negative; *p* = 0.036) (Figure 7H). Thus, together, these findings suggest that retention of Chr13-encoded *miR15a/miR16-1* is associated with restraint of p53 expression during cycling, likely reflecting feedback circuitry between p53 and *miR15/mi16* [58].

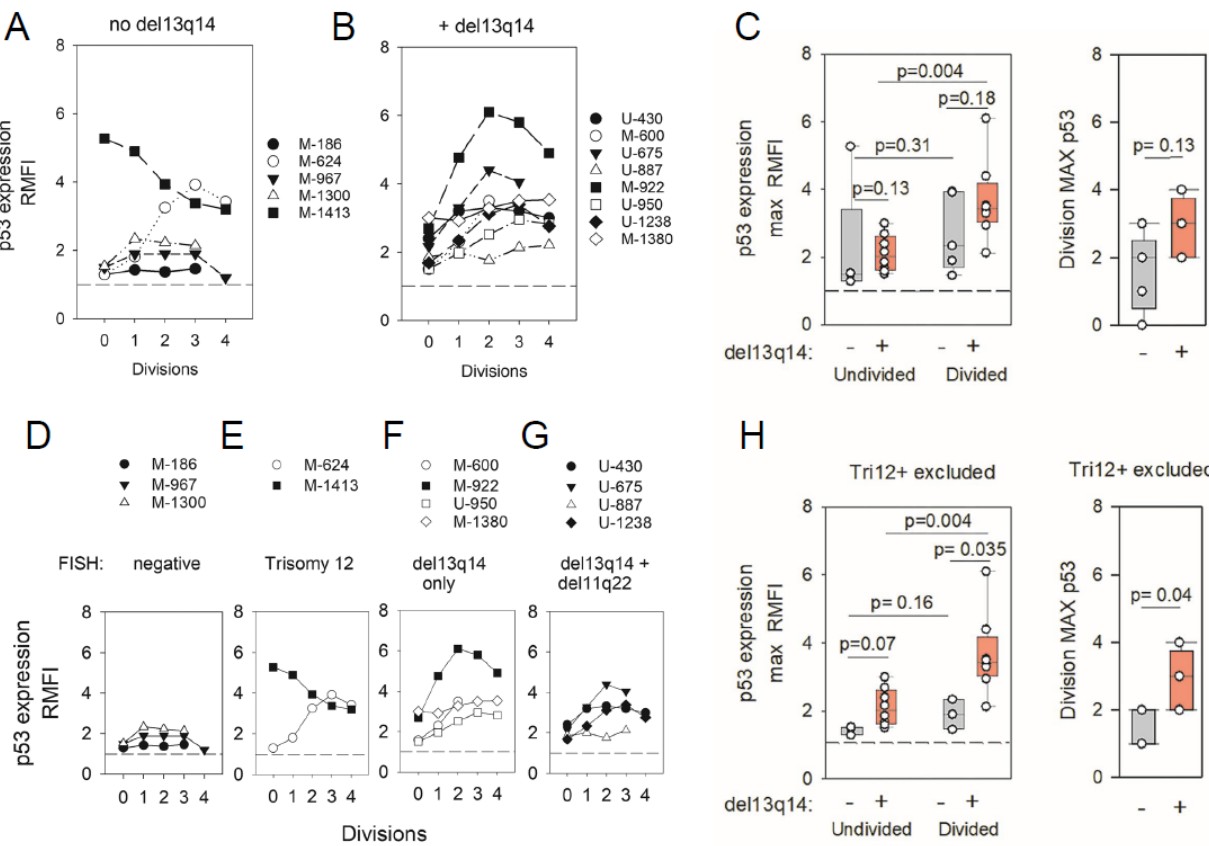

**Figure 7.** B-CLL populations with del(13q) manifest heightened p53 protein during cycling. (**A**) del(13q)−/− CLL or (**B**) del(13q)+ CLL cohorts were assessed for intracellular p53 protein during ODN + IL15-triggered clonal expansion. Histograms indicate p53-PE (RMFI) within blasts of varying division history. (**C**) Box plots comparing p53 levels within undivided and divided cells within del(13q)−/− and del(13q)+ CLL cohorts. (**D**–**G**) Plots of p53 expression versus division frequency in ODN + IL15-stimulated CLL populations, segregated on the basis of FISH-determined chromosomal anomalies (Table 1). (**H**) Box plot analysis of p53 levels (as in (**C**)) with the exclusion of Tri12+ CLL from analysis. Statistical analyses involved a paired, 2-sided *t*-test (comparisons of undivided versus divided cells of the same cohort) and an unpaired, 2-sided *t*-test (inter-cohort comparisons). When non-parametric p53 data distribution was detected in the del13−/− cohort (**C**), statistical evaluations involving the latter were performed with the Mann–Whitney rank sum test.

## 3. Discussion

This study's observations that BCL2 protein progressively declines during active CLL clonal expansion in a manner that is paralleled by progressively lesser vulnerability to venetoclax-induced apoptosis might be clinically important. First, these findings support the view that CLL is challenging to treat, at least in part due to its dimorphic nature, characterized by quiescent cells in blood and small foci of proliferating cells in lymphoid tissues, each with differing metabolic requirements and mechanisms for survival. Second, they emphasize that a cure for CLL will require the elimination of CLL cells within both compartments. While negligible in terms of overall leukemic cell burden, the cycling cell subset is nonetheless clinically relevant [67–69].

One important attribute of actively cycling CLL cells, shared by cycling normal B lymphocytes, is a significant rise in mutagenic enzyme AICDA [30,62,70]. Increases in AICDA protein levels and AICDA function are accentuated in cells with the most divisions, both in normal and CLL B cells [30,62,70,71]. Thus, based on this study's findings, we consider it probable that during venetoclax treatment of patients, the highly divided CLL cells with the greatest capacity for generating AIDCA-driven mutations will preferentially

survive. Once these actively cycling cells later revert to relatively quiescent circulating cells with upregulated BCL2 [72,73], analogously to the transition of germinal center B cells into memory cells [74], any clonal variants with functional *BCL2* gene mutations that impair venetoclax binding to BCL2 [14,75] will be venetoclax resistant and could eventually become the dominant CLL subclone [30,70,71]. This hypothesis is consistent with evidence that BCL2 mutations in CLL clonal populations occur prior to treatment and are *acquired* during treatment [14,15,18]. A direct correlation between AICDA expression and BCL2 mutations is not yet demonstrated in CLL, to our knowledge. However, such a correlation has been made in follicular lymphoma [76] in which the *BCL2* gene is characteristically translocated to the IGH locus. With this t(14;18) translocation, the *BCL2* gene comes under the strong influence of the IGH 3' super-enhancer that heightens AICDA targeting/function to this locus [77,78].

The study's evidence that pharmacological inhibition of survivin (BIRC5)—a protein that directly blocks caspase 9 activation [79–82] with a separate role in promoting cell mitosis [81,83]—is effective at both inducing apoptosis and blocking further CLL divisions suggests that survivin inhibitors could be a beneficial adjunct with venetoclax. One notable advantage of using combined venetoclax and survivin inhibitor as a first-line treatment for CLL patients is that the former will debulk patients of circulating and quiescent tissue CLL cells, while the latter should mitigate the emergence of clonal variants with deleterious mutations, including those in *BCL2* and *TP53*. Patients predicted to benefit most from this approach are those demonstrating signs of CLL proliferation in lymphoid tissues, e.g., lymphadenopathy and/or elevated blood frequency of survivin-positive leukemic cells [21] and/or CLL cells with an elevated CD5/CXCR4 ratio [73,84,85]. Both of the latter are indicators of recent cycling [73,84,86]. As recently reviewed [87], among adjunct approaches currently in clinical use to enhance venetoclax efficacy in first-line treatment of CLL, the most effective are second-generation covalent BTK inhibitors, acalabrutinib, and zanabrutinib, which in combination with venetoclax, yield patient 4-year survival rates of 88% and 94%, respectively [87]. Unfortunately, there is still not a demonstrated cure for CLL, and the necessary chronic exposure to BTK inhibitors can elicit adverse events, e.g., major cardiac or bleeding disorders [87,88]. The fact that survivin protein is characteristically absent in quiescent cells but upregulated in proliferating cells [48] is important for possible employment of survivin inhibitors as an adjunct to venetoclax. However, a caveat to targeting survivin is the latter protein's important role in furthering the cycling of precursor cells during B cell development [83] and physiologically relevant cycling of all cell lineages [48]. That said, such effects should cease with intermittent use of specific survivin inhibitors. It warrants noting that YM155 has demonstrated anti-tumor efficacy in pre-clinical in vitro studies and xenograft mouse models of diffuse large B cell lymphoma [89,90] and B cell non-Hodgkin lymphoma (NHL), particularly in combination with other agents [36]. In addition, a recent multicenter phase II study of combined YM155 plus rituximab treatment of patients with relapsed aggressive B cell NHL demonstrated tolerance/efficacy [91].

This study's finding that activated, cycling CLL cells manifest a significant decline in BCL2 protein might be seen as surprising, given in light of other reports that, while not elevated, BCL2 protein did not decline following CLL activation in culture [28,29]. However, this disparity likely reflects several factors: the relatively short duration of past cultures (24 to 72 h) [28,29]; BCL2 is a relatively long-lived protein [53]; and undivided and divided blasts were not distinguished [28,29]. In more extended cultures of CLL cells activated by CD40L-expressing stromal cells and IL-4, reduced BCL2 protein was observed, but the effect of cycling was not examined [92].

The study's effort to illuminate mechanism(s) for BCL2 repression during dynamic CLL clonal expansion points to the contribution of an IL15 → STAT5/PI-3K pathway essential for CLL growth [35]. The latter fosters elevated p53 TF that, based on other molecular studies, should drive p53-driven transcriptional activation of *miR15a/miR16-1* [50], two critical BCL2-repressive miR encoded on Chr13 [9,50]. The absence of experiments directly

monitoring miR levels precludes a definitive conclusion, but the above premise is supported by observations that BCL2 protein levels within cycling del(13q)+ CLL populations (deleted *miR15a/miR16-1*) are greater than within cycling del(13q)−/− CLL. Providing further support for p53-driven BCL2 repression is the study's finding that the important IL15/STAT5/PI-3K signaling axis [34,35], as shown here, leads both to pre-cycling increases in p53 TF and diminished *BCL2* mRNA. It is of interest that a p53 pathway for BCL2 repression may also affect the growth of non-transformed B cells. Human follicular B cells stimulated by BCR:CD21L, IL4, and BAFF manifest early and sustained increases in p53 accompanied by progressively declining BCL2 protein during successive divisions [32,93]. Furthermore, and importantly, a recent report indicates that within human/mouse germinal centers (sites of extensive B cell growth and related DNA damage [60] where p53 protein accumulates [60]), the levels of *miR15/miR16* rise and BCL2 declines, as compared to levels in quiescent follicular B cells [94]. In the same study, Cre recombinase-mediated selective deletion of *miR15/miR16* in GC B cells precipitated a significant rise in BCL2 protein [94].

Notwithstanding, factors in addition to p53 → *miR15a/miR16-1* driven BCL2 repression may also contribute to the progressive decline in BCL2 protein during cycling. This was suggested by the observation that del(13q)+ and del(13q)−/− CLL cohorts exhibit a similar proportional decline in BCL2 levels as undivided cells progress to 3 divisions. Serial dilution of pre-existing long-lived BCL2 protein with successive divisions, in the context of reduced synthesis, might be a contributing factor. Also, p53 TF is reported to lower *BCL2* synthesis by blocking the binding of POU4F1 TF to the *BCL2* promotor [95]. Furthermore, *miR-125b* and *miR155* are reported repressors of BCL2 in CLL cells activated in vitro by CD40L-stromal cells and IL4 [96]. Effects of p53 TF transactivation of an analogous *miR15b/miR16-2* cluster on Chr3 [58] are not excluded. While the GC-specific BCL6 TF directly represses *BCL2* transcription within GC B cells [31,58], its contribution seems unlikely because histological studies show that CLL pseudofollicles, in contrast to germinal centers, express low BCL2 and no BCL6 protein [22,97]. Moreover, when xenografting primary CLL cells and activated autologous T cells into alymphoid mice, BCL6 protein was not found in proliferating CLL cells [98]. Further studies are needed to better understand of how these, and possibly other [4,99] mechanisms, integrate in repressing BCL2 during CLL cycling. Considered quite unlikely is the possibility that diminished BCL2 expression during CLL cycling represents selectively greater cycling of clonal variants bearing *BCL2* mutations, which diminish BCL2 expression. First, the BCL2 decline was uniformly noted in all 14 CLL clonal populations evaluated, and the likelihood of each having such a rare background mutation in CLL [13,14] is improbable. Second, none of the patients had received earlier venetoclax treatment, which is linked to the acquisition of clonal variants with *BCL2* mutations [13–15].

In the context of the observed BCL2 protein loss during cycling, the significant evidence that BCL2 functions as a brake on cell cycle progression at the G1 → S phase transition, in addition to its direct pro-survival role [100–102], requires emphasis. This growth-suppressing function may explain why mechanisms to downregulate BCL2 during active lymphocyte growth have evolved [31,32,93,94] and are retained in certain B cell malignancies. Interestingly, while prior histological studies show an inverse relationship between Ki67 and BCL2 in some follicular lymphomas (FL), marginal zone lymphoma, and a subset of CLL/small lymphocytic lymphoma (SLL) [103], this inverse relationship did not apply to more aggressive mantle zone lymphoma, diffuse large B cell lymphoma (DLBCL) and certain other CLL/SLL [103]. Rather, these latter tumors retained high BCL2 expression in Ki67+ proliferating cells [103], which in both aggressive FL and DLBCL is explained by *BCL2* gene translocation to the Ig heavy chain locus [104]. The finding of elevated BCL2 protein in aggressive B cell malignancies suggests that mechanisms for BCL2 repression and BCL2 cell cycle control are no longer operative. It may be relevant that while both FL and DLBCL are typified by a high frequency of AICDA-driven BCL2 mutations linked to aggressive growth [105–107] when examined, only a fraction resulted in augmented BCL2 pro-survival function [107]. While the assumption was that these other mutations were

non-functional, it remains possible that they diminished the growth-attenuating function of BCL2.

Our study's additional discovery that p53 TF protein levels during cycling were significantly greater within the del(13q)+ cohort than in the cohort devoid of chromosomal anomalies is fully consistent with past molecular evidence that Chr13-encoded *miR15a/miR16-1* exert negative feedback on *TP53* transcription [50,58]. Both the latter, as well as *miR155*-mediated suppression of *TP53* transcription [108,109], may explain why *TP53* mRNA levels decline within ODN + IL15 activated CLL cells. Mechanisms for feedback repression of p53 TF are undoubtedly important during stressful B cell clonal expansion because the p53 protein elicits cell cycle blocks and promotes apoptosis [110]. Without its fine-tuned regulation, the expansion of individual B cells that encountered antigen and other growth stimuli would cease.

Some discussion is justified regarding the aberrantly elevated levels of p53 protein noted in two ODN + IL15-activated CLL with Tri12+ (despite their del(13q)−/− status). While *TP53* mutations alone and in combination with del(17p)+ have been linked to elevated levels of abnormal p53 protein, neither are likely explanations. First, del(17p) was absent, and, furthermore, both *TP53* mutations and del(17p) are significantly underrepresented in Tri12+ CLL, as compared to other CLL populations [111,112]. As a reasonable premise, we suggest that negative feedback of *miR15a/miR16-1* on p53 protein *synthesis* is muted within clonal populations exhibiting particularly robust cycling and accompanying DNA modifications that foster ATM activation and the latter's ensuing *stabilization* of p53 protein [32,113]. Consistent with this interpretation, past assessments of the extent of cycling observed within the above and other Tri12+ clones showed that Tri12+ CLL populations, in general, exhibit a particularly vigorous growth response to ODN + IL15 (within reference [2], see both Fig. 8C and Supplemental Fig. 3). The *NOTCH1* mutations present with relatively high frequency in Tri12+ CLL [111,114] might contribute to the robust growth of Tri12+ CLL clones upon receiving appropriate stimulation. An increased gene dosage effect of Tri12+ might be a further influencing factor, particularly for Chr12-encoded insulin-like growth factor I [115]. In support, IGF1R signaling augments CLL cell survival [116], and in CLL, elevated IGFR1 levels are associated with Tri12+, NOTCH1 mutation, and aggressive clinical course [117]. Finally, the enriched culture medium used in our studies of normal and CLL B cell growth is supplemented with added insulin [2,118] that, like IGF1, can signal cells via IGF1R [119]. While a gene dose effect of Tri12+ on MDM2 expression could influence maximal p53 levels via MDM2-mediated degradation of p53, a past study showed no difference in baseline MDM2 protein levels within Tri12+ and CLL without this aberration [120].

Given this study's mechanistic insights, it warrants considering the functional implications of del(13q) emergence as an early driver mutation in ≥50% of the cases of monoclonal B cell lymphocytosis and ensuing CLL [121]. The resulting loss of *miR15a/miR16-1* should alter the mutant cell's response to activating stimuli in its tissue environment that stabilize the p53 protein [32] in diverse ways. First, the synthesis of p53-driven *miR15a/miR16-2* [58], which suppresses BCL2 mRNA/protein, would be dampened, resulting in elevated BCL2 protein that both promotes B cell viability and restricts growth. Furthermore, with the deletion of the *miR15a/miR16-1* locus, a means of providing negative feedback on p53 expression [58] is also lost. Thus, the clone might be expected to exhibit restrained growth but elevated survival unless the CD5+ clone acquires additional genetic aberrations to counteract the compromised growth/survival, which follows the significant stabilization of p53 protein during vigorous cell growth. While speculative, the above hypothesis is consistent with the typically slow progression of monoclonal B cell lymphocytosis (MBL) [122,123]; high prevalence of del(13q) in early MBL [124,125]; the better clinical outcome of CLL with del(13q) as the sole chromosomal anomaly [126]; and the progressive appearance of genetic anomalies that alter expression and/or function of p53, either directly (e.g., TP53, ATM, or MDM2) or indirectly (e.g., NOTCH1, SF3B1, BIRC3, and RPS15 [114,125,127–136].

## 4. Materials and Methods

### 4.1. CLL Patient Samples

B cell CLL specimens were obtained from peripheral blood (PB) with the exception of M-CLL967, isolated from a therapeutically removed spleen; their earlier genetic characterization (Table 1) and further characterization of patient blood were presented earlier [2,34,35]. The selection of CLL for this study was based on past evidence of significant cycling in cultures activated by synergistic ODN + IL15 [2,34,35]. None of the CLL specimens were from patients with prior exposure to venetoclax, indicating that the cohort under study does not contain CLL populations with a heightened potential for expansion of BCL2 mutant subclones that are characteristically acquired during venetoclax treatment but not present before treatment [13,14,19]. Furthermore, within this study, only a minority of CLL donors had received any prior treatments: (a) 2/14 evaluated for BCL2 and MCL1 (U-CLL430 and U-CLL675); (b) 2/10 evaluated for sensitivity to venetoclax or YM155 (U-CLL675 and M-CLL849); and (c) 4/14 evaluated for p53 expression (U-CLL430, U-CLL675, M-CLL849, and U-CLL950). Thus, the employed cohort has only a minority of CLL populations with the possible treatment-facilitated amplification of pre-existing or acquired mutations in molecules influencing its expression/function [19], including *TP53* (p53).

### 4.2. CLL Cell Isolation from Patient Blood (n = 29 CLL) and Spleen (n = 1 CLL)

As described [2], CLL cells were selected from peripheral blood by Ficoll separation followed by negative selection using RosetteSep Human B Cell Enrichment Cocktail (Stemcell Technologies; Cambridge, MA, USA). Exceptions were (a) M-CLL1300, frozen as Ficoll-isolated PBMC, and (b) M-CLL967 from therapeutically removed spleen. The latter was frozen as Ficoll-isolated cells and, upon recovery, selected for high-density (quiescent) cells by 40–75% Percoll gradient centrifugation followed by cell collection at the 55/75% interface [137].

### 4.3. Purity of CLL Populations

Information regarding the frequency of CD19 + CD5+ CLL cells and other contaminating leukocytes in the original blood sample and purified CLL populations was provided in an earlier report [2]. Surface marker expression on clonal populations in this study was re-examined in some CLL following recovery of viable cells after defrosting, both prior to culture and/or after 5–6 d of ODN + IL15 activation (Supplementary Figures S1–S3). Those subjected to the B cell enrichment step showed negligible, if any, contamination with non-CLL cells at t = 0 (Supplementary Figure S1); furthermore, following 6 d culture with activating ODN + IL15, the vast majority of proliferating cells possessed an unambiguous CLL phenotype (CD19 + CD5+) (Supplementary Figure S2). However, Supplementary Figure S3 shows that, in the case of M-CLL1300 PBMC (sole PBMC population without the B cell purification step), minor contamination with non-B cells prior to culture led to atypical d6 cycling in IL15-only cultures, not seen in purified CLL populations [2]. Cycling elicited by exposure to IL15 alone was due to CD19-negative cells and likely represented initial contamination of the above PBMC with T and NK cells, two non-B cell populations that each cycle extensively with solely IL15 exposure [138,139]. Within cultures with both ODN + IL15 (synergistic for growth of CLL B cells [2,34,35], both CD19-negative and CD19+ cells divided, with CD19+ dominating those with 0 to 3 divisions and CD19-negative dominating those with 4 to >6 divisions. Therefore, in assessing BCL2/MCL1/p53 protein expression within CLL1300 cells following ODN + IL15 stimulation, only data from CFSE-gated fraction with 0 to 3 divisions is presented, with this qualification. The additional possibility that very minor contamination with residual normal B cells (NRB) contributes significantly to the ODN + IL15-elicited progeny is highly unlikely for the following reasons. First, only normal memory B cells (but not naïve B cells) proliferate upon culture with ODN + IL15 without further antigen receptor signaling [140], and although memory B cell number (cells/mL of blood) within CLL patients is similar to those observed in normal aged individuals [141], memory B cell numbers in both considerably

decline with aging [142]. Thus, the number of NRB potentially responding to ODN + IL15 is exceedingly low compared to the very dominant CLL population. Second, all CLL cultures were monitored daily within the first 3 days of culture, and often longer. Purified CLL populations, which later showed notable cycling, typically manifest relatively uniform cell enlargement early in culture and the formation of clusters of enlarged cells by d3. If memory NRB were responsible for the noted cycling, only a limited number of cells in the culture would have demonstrated these early changes. Third, when tested, cycling cells in ODN + IL15-stimulated purified CLL cells are CD19+ CD5+; CD3+ T cells and CD16+ NK cells were not detected (Supplementary Figure S2); Fourth, a past study [143] involving CLL cells purified similarly to this study, found that dividing cells within d7 cultures stimulated by ODN + CD40L + cytokines, exhibited the same IGV clonal rearrangement as the starting population. Fifth, a recent study on the NRB population within the blood of multiple CLL patients by single-cell (sc)-RNA sequencing and sc-VDH sequencing [144] found that the NRB population is significantly populated by subclones linked to the major CLL clone [144].

While an approach to eliminating possible contributions of NRB or other contaminants to the cycling population might involve selective monitoring of intracellular protein expression within gated CD19 + CD5+ cells, this was not employed for two reasons: (a) some CLL manifest less CD5 than others, with a portion of the clone not exceeding CD5 expression above background, and (b) surface staining, followed by the fixation/permeabilization steps needed for intracellular staining, reduces surface antigen staining intensity [145] (unpublished findings, P. Mongini). Both factors mean that the later gating approach would inappropriately exclude a fraction of cycling CLL cells.

### 4.4. Culture Conditions

To monitor cell divisions, CLL clonal populations were pre-labeled with CFSE [2] and cultured for 5–7 days in an enriched medium optimized for B cell viability [93,118], at $10^5$ cells/200 μL in 96-well culture plates, with synergistic activating stimuli [2,34,35]: CpG DNA TLR-9 ligand (ODN-2006; Invivogen, San Diego, CA, USA; final concentration of 0.2 mM) and recombinant human IL15 (R&D Systems, Minneapolis, MN, USA; final concentration of 15 ng/mL). Parallel cultures contained IL15 alone and, occasionally, ODN alone or medium alone. Of note, earlier studies [2] had determined that although the CLL clonal populations selected for these experiments exhibit significant ODN + IL15-induced cycling by d5–7 of culture, the CLL cells remained largely quiescent when cultured with IL15 alone. Culture with ODN alone uniformly resulted in CLL cell enlargement, but depending on the IGHV subset of the clone, it resulted in either prominent apoptosis (M-CLL) or minimal cycling (U-CLL) [2]. Both M-CLL and U-CLL cells showed marked synergy between ODN and IL15 in eliciting protracted cycling [2].

For experiments assessing the impact of inhibitors of venetoclax and survivin on CLL growth and viability, determinations of absolute viable cell yield were made by pulsing triplicate cultures just prior to harvest, with a known number of fluorescent standardizing beads (CountBright absolute counting beads (Molecular Probes, Life Technologies, Waltham, MA, USA), followed by fixation and flow cytometric analysis involving gating of viable/dead intact cells by V450-PB dye exclusion and SSC, as earlier described [2,34]. Inhibitors were pulsed into d4(d5) cultures for a period of 36–48 h before harvesting cultures with added beads.

### 4.5. Inhibitors

Pharmacologic inhibitors were reconstituted in DMSO and stored at −80 °C in aliquots prior to dilution in a culture medium for use. BCL2 inhibitor, venetoclax (ABT-199; Selleckchem, Houston, TX, USA; cat #S8048), was used at final culture concentrations of 50 to 1.9 nM; survivin inhibitor, YM155 (Calbiochem, San Diego, CA, USA; cat #574662), was tested at 1000 to 37 nM. Each of the above was added to cultures soon after ODN + IL15-activated cultures (Table 1) began a burst of rapid cycling [2]. In other cultures involving

inhibitors of STAT5 and PI-3K, LY294002 (Selleckchem), an inhibitor of PI3K p110α, β, and ɣ isoforms [146], was used at 20 mM; STAT5 inhibitor, pimozide (also known as STAT Inhibitor III) (Calbiochem; CAS2062-78-4) and STAT5 Inhibitor II (CAS 285986-31-4) were used at doses near their reported IC50 values: pimozide (5 mM) [147] and weaker STAT5 Inhibitor II (47 mM) [148].

### 4.6. Intracellular Staining for Cytoplasmic BCL2, MCL1, and Survivin Proteins

Following culture harvest, cells were washed in cold PBS; permeabilized in Cytofix/-Cytoperm buffer (BD Biosciences, San Jose, CA, USA), followed by washes in Perm-Wash buffer (BD); and incubated for 35 min (RT) with primary specific Ab (or IgG control), followed by washing and 30 min exposure to secondary Ab (PE-labeled goat F(ab')2 anti-mouse IgG (H&L) absorbed with human Ig; Southern Biotech, Birmingham, AL, USA; #103209). Each staining was performed in duplicate, and washed cells were refixed with 2% EM-grade formaldehyde prior to flow cytometry. Primary mAbs used: anti-BCL2 (clone 124 specific for aa 41–54; Dako, Santa Clara, CA, USA; cat #M0887); anti-MCL1 (clone 22; BD Pharmingen or clone RC13; Santa Cruz Biotechnology, Paso Robles, CA, USA; cat# sc-56152); and anti-survivin (clone D8; Santa Cruz Biotech cat #sc-17779). This protocol detects cytoplasmic survivin as the primary site of survivin in CLL cells [149] but not nuclear survivin [150] or survivin expressed on cell membranes [151,152].

### 4.7. Staining for Nuclear p53 Protein

p53 was monitored using a p53-staining kit (PE-anti-p53, clone DO7, and PE-IgG control mAb; BD Pharmingen, San Diego, CA, USA; cat# 556534), as described earlier [32]. Briefly, harvested cells were fixed with EM-grade formaldehyde (3%) for 10 min, followed by 30 min permeabilization with ice-cold methanol (90%), washing, and intracellular staining.

### 4.8. Flow Cytometric (FACS) Analyses

Two-color flow cytometry was employed for assessing intracellular protein levels within CFSE-labeled cells of differing division status and involved either Fortessa or LSR II flow cytometers (BD Biosciences), followed by data analysis with FlowJo software (Version 10.1r1). In analyses for BCL2, MCL1, survivin, and p53 protein, stained/fixed cells were gated for viable cells on the basis of light scatter (FSC/SSC) or by plotting SSC versus fluorescence of viability-assessing dye, V450-Pacific blue (V450-PB). Levels of intracellular BCL2, MCL1, and p53 were determined by calculating RMFI (ratio of the geometric mean of specific protein fluorescence to the geometric mean of IgG Ctrl fluorescence) for CFSE-gated subsets of duplicate samples.

### 4.9. Immunoblotting for p53 Protein

Blotting experiments with validated anti-p53 mAb (DO-1, which detects p53 isoforms) and anti-β-actin mAb were performed on SDS-PAGE separated lysates from d0 or d5 CLL cells cultured at $10^6$ cells per well 24-well plates, using techniques described for assessing p53 in activated human B cells [32].

### 4.10. Quantitative Assessment of BCL2, MCL1, and TP53 mRNA by Reverse Transcription Real-Time PCR (q-RT-PCR)

For brevity, q-RT-PCR is hereafter referred to as qPCR. Total RNA was isolated from 1 to 2 million cells using Mini-prep Qiagenkit (Qiagen, Gaithersburg, MD, USA); cDNA prepared with oligo(dT) primers; and specific cDNA amplified in triplicate using specific probes, as described in our earlier studies [32,34,62]. For specific amplification by TaqMan QPCR (Applied Biosystems, Foster City, CA, USA), 2.5 μM specific probe (from human Universal Probe Library of Roche Applied Science [Indianapolis, IN, USA]), together with intron spanning, optimized forward (F) and reverse (R) primers (each at 10 μM) (ProbeFinder version 2.50 for human [Roche Diagnostics]) were employed;

assays were performed in triplicate. Probe accession numbers and primer sequences were obtained from the RefSeq database (https://www.ncbi.nlm.nih.gov/refseq/). *BCL2* and *MCL1* analyses: *GAPDH* was used as endogenous control for the calculation of ΔCt values and involved the following probes and primers: *GAPDH*: (Universal Probe Library probe 60; accession no. NM_002046.3) with primers, F = 5′-agccacatcgctcagacac-3′ and R = 5′-gcccaatacgaccaaatcc-3′ (synthesized by Eurofins mwg\operon). *BCL2*: probe #75 (accession no. NM_000633.2) with primers, F = 5′-agtacctgaaccggcacct-3′ and R = 5′-gccgtacagttccacaaagg-3′. *MCL1*: probe #4 (accession no. NM_021960.4) with primers, F = 5′-aagccaatgggcaggtct-3′ and R = 5′-tgtccagtttccgaagcat-3′. *TP53* (p53) analyses: probe #12 (accession no. NM_001126114.1) with primers, F = 5′-aggccttggaactcaaggat-3′ and R = 5′-cccttttttggacttcaggtg-3′ were used, together with the endogenous reference control, β-actin (*ACTB*), probe #64 (accession no. NM_001101.3) and primers, F = 5′-ccaaccgcgagaagatga-3′ and R = 5′-ccagaggcgtacagggatag-3′. ΔCt values were determined by comparing cycle amplification values for the detection of specific mRNA versus cell reference control (GAPDH) mRNA (note: greater ΔCt values represent lesser *BCL2* or *MCL1* mRNA). Fold change was calculated by comparing ΔCt values of treated versus untreated groups, using the $2^{-\Delta\Delta CT}$ method [66] with analysis by RQ Manager 1.2 (Applied Biosystems).

### 4.11. Phase Microscopy

Photographs were taken at a magnification of 200× using phase microscopy (Olympus BX40 phase-contrast microscope and Olympus DP20 camera).

### 4.12. Statistics

Tests for determining statistical significance are indicated within figure legends. Typically, a two-sided *t*-test was employed for normally distributed data: paired if involving values from the same CLL populations and unpaired if comparisons involved differing CLL populations. An unpaired *t*-test was also employed when data were normalized by providing a value of 1 to control cultures without treatment. In cases where data distribution did not pass the Shapiro–Wilk normality test, the nonparametric Mann–Whitney rank sum test was used. Box plots were used to show summed statistics for various cohorts of CLL studied. In box plots, the upper part of the box represents the third quartile (75th percentile) and the lower part of the box the first quartile (25th percentile); upper and lower whiskers represent error bars for the 10th and 90th percentiles, respectively; outliers are shown as individual points. Furthermore, median values for the grouped data are shown by solid lines within each box. Statistical significance was determined when $p \leq 0.05$; determinations were made with either Sigma-Plot 13 or Excel (Microsoft 365 MSO; Version 2402).

### 4.13. Study Approval

The studies were approved by the Institutional Review Board of Northwell Health (08-202A). Before blood collection, written informed consent from patients was obtained in accordance with the Declaration of Helsinki.

**Supplementary Materials:** The following supporting information can be downloaded at https://www.mdpi.com/article/10.3390/lymphatics2020005/s1.

**Author Contributions:** H.L. performed culture and staining experiments involving BCL2, MCL1, and p53 and helped with the manuscript; S.H. performed RNA isolation and qPCR of *TP53* transcripts and helped with the manuscript; and R.G. performed experiments involving RNA isolation and qPCR of *BCL2* and *MCL1* transcripts as well as those involving venetoclax and YM155. K.R., S.L.A. and J.E.K. provided characterized CLL blood specimens and helped with the manuscript. N.C. led the acquisition, *IGHV* analysis, and initial assessments of purity in CLL specimens used for this study and helped with the manuscript. P.K.A.M. conceived experiments, performed certain functional experiments, analyzed data, and wrote the manuscript. All authors have read and agreed to the published version of the manuscript.

**Funding:** Karches Family Foundation and NIH grants (N.C.: CA081554 and P.K.A.M.: AR061653).

**Institutional Review Board Statement:** The studies were approved by the Institutional Review Board of Northwell Health (08-202A). Before blood collection, written informed consent from patients was obtained in accordance with the Declaration of Helsinki.

**Data Availability Statement:** Underlying data are available by correspondence with the contributing author upon request.

**Acknowledgments:** The authors would like to acknowledge the technical contributions of Joshua Trott and Jennifer Nieto to early experiments.

**Conflicts of Interest:** The authors have declared that no conflicts of interest.

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
