# Peer review of "BCL2 Protein Progressively Declines during Robust CLL Clonal Expansion: Potential Impact on Venetoclax Clinical Efficacy and Insights on Mechanism"

_2813-3307, doi:10.3390/lymphatics2020005_

Round 1

Reviewer 1 Report

Comments and Suggestions for Authors

-       Figure 1B: The difference between FigA and FigB in term of experimental condition is not well describe in the text and is confusing. In Fig A, the authors compare BCL2 expression at day=0 and day=5 after stimulation. In Fig B, the authors compare BCL2 expression in primary CLL cells stimulated with ODN+IL15 vs unstimulated cells after 6 days. It is very surprising that primary CLL cells can stand 6 days only with media as they usually die quickly without stimulation or co-culture. This need to be consolidated with more replicates. I would gate on CD19 and CD5 expression before BCL2. It would be also interesting to show MCL1 expression as well. 

-       Fig D and E are not histograms (as mentioned in the text and the legend) but curves of evolution of BCL2 in subset of divided cells. The authors should show if the reduction of BCL2 expression is significant as they did in figure 1F and 1G, by comparing the RMFI of each subset of division to day 0. 

-       The term of “blast” is not appropriate for cycling CLL cells. It should be replace throughout the text. 

-       Fig1G : at 2 division, p value is 0.03 while it is p=0.04 in the text.

Overall,   the diminution of BCL2 expression is not significant and is not clearly demonstrated here. However, all the paper relay on this observation. 

-      Fig2A : The level of BCL2 and MCL1 mRNA needs to be show at d=0 before the 20h priming with ODN to have an idea of the basal level of their expression. 

-       Fig2A and 2B : only 2 replicates at 4h after IL15. At 20-28h, it seems that both BCL2 and MCL1 mRNA expression slightly increase as the DeltaCt value is higher for each patient in the condition with IL15 treatment vs without (medial alone). Therefore, in Fig C, the calculation of the DeltaDeltaCt (fold change ) is not correct :  it should be greater to 1 and not below at 20-28h. I question the general conclusion of this figure. 

-       The expression of the results in Fig2C is not very intuitive. Could the authors show how the DeltaCt vary (or the fold change) with each inhibitor in comparison with untrated IL-15 pulsed cells ?

-       Fig3A B E and F are not histograms, as it is mentioned in the text/legend

-       In this figure, the authors show that there is less repression of BCL2 during cycling in del13q cells, while MCL1 doesn’t change.  They suggest that the loss of mir15/16a contribute to maintain a higher level of BCL2. However, there is no demonstration of such mechanism. 

-       Figure 4: Experiments are done with an insufficient number of replicates in this part of the study

-       Fig4A : what is the rational to compare stimulation with IL15 only vs IL15+ODN? IL15 stimulation has not been investigated so far in the study to look for BCL2 protein or mRNA expression. Wouldn’t it be more relevant to compare cells in media alone or primed with ODN +/- IL15 ? Those are the conditions used in figure 2A.

-       These experiments really need to be done on replicates to see if the increase of p53 is significant. 

-       Same remark for the interpretation of the TP53 qPCR analysis in figure 4E as in Figure 2A-C. Level of expression of TP53 mRNA increase when stimulated with ODN+IL15. The fold change should be greater than 1, but not sure is significant. Therefore I disagree with the conclusion of this Figure. 

-       The interpretation of the p53 data should also be done with regard of the TP53 mutation status, not only with the del17p status, as 30% of isolated mutation occur in CLL patients. 

-       In Figure 5, the authors show that p53 level is higher in del13q dividing cells as compared to del13 negative cells. They suggest that mir15/16 contribute to p53 activation. However, the increase of p53 expression is not significant in divided cells as compared to undivided cells. 

-       The difference of level of expression of p53 among patient cannot be explain or attributed only by the status of the other cytogenetic aberration, such as tri12, specially because the number of samples are too low to draw any conclusion. This is more relevant to analyze the data with regard of the TP53 mutation status. 

- Figure 6 : The reduction of sensitivity to venetoclax in cycling cells is explain by a mechanism of BCL2 repression that is not clearly demonstrated earlier in the paper. 

Author Response

The authors thank this reviewer for time and thought placed in review of our manuscript.  We feel we have addressed the suggestions made and all of these are detailed in the attached PDF.   Further understanding of the significant changes will require inspection of the substantially revised manuscript.

Thank you again.

Reviewer 2 Report

Comments and Suggestions for Authors

All the experiments are sustained in the hypothesis that del 13q14 containing important microRNAs that modulate apoptosis is lost in some patients studied, and this loss downmodulates BCL2 expression. However, they did not measure miR15a and miR16-1 to prove it. Also, sequencing of BCL2 will be important to solve the question regarding if the down-regulation of BCL2 is due to miR or mutations occurring specifically at the BCL2 gene. 

After IL15 stimulation, why are they measuring STAT5 as an output, as in B cells usually JAK1-STAT3 is activated instead of STAT5.

BCL2 is one of the targets in CLL. However, many other targets and mutations exist, especially in mRNA-controlling genes. Explain how these results could be interpreted in the light of the other driver mutations (Landau et al., 2015.  DOI: 10.1038/nature15395.) as in the introduction and discussion are focused on apoptosis resistance, but other hypotheses and clones exist.

Discuss how other clones could resist BCL2 inhibitor treatment (Shadman, 2023 DOI: 10.1001/jama.2023.1946) and how this work will help a specific cell population and patients.

Please uniform the text with black letters. There are gray and black marks throughout the text.

Please review references and citations as on page 16 line 460 (Croce et al is 54 not 53.)

Comments on the Quality of English Language

A minor English review is necessary.

Author Response

The authors thank this reviewer for time and consideration given in review of our earlier submitted manuscript.   We feel we have addressed the reviewer's comments and our response is detailed in the attached PDF.   Full understanding of the significant changes made will require inspection of the substantially revised manuscript.

Thank you again.

Round 2

Reviewer 1 Report

Comments and Suggestions for Authors

The authors have answered the concerned about the manuscript.